# ATP biosensor reveals microbial energetic dynamics and facilitates bioproduction

Xinyue Mu[1], Trent D. Evans[1] & Fuzhong Zhang [1,2,3] ✉

Adenosine-5′-triphosphate (ATP), the primary energy currency in cellular processes, drives metabolic activities and biosynthesis. Despite its importance, understanding intracellular ATP dynamics' impact on bioproduction and exploiting it for enhanced bioproduction remains largely unexplored. Here, we harness an ATP biosensor to dissect ATP dynamics across different growth phases and carbon sources in multiple microbial strains. We find transient ATP accumulations during the transition from exponential to stationary growth phases in various conditions, coinciding with fatty acid (FA) and polyhydroxyalkanoate (PHA) production in *Escherichia coli* and *Pseudomonas putida*, respectively. We identify carbon sources (acetate for *E. coli*, oleate for *P. putida*) that elevate steady-state ATP levels and boost FA and PHA production. Moreover, we employ ATP dynamics as a diagnostic tool to assess metabolic burden, revealing bottlenecks that limit limonene bioproduction. Our results not only elucidate the relationship between ATP dynamics and bioproduction but also showcase its value in enhancing bioproduction in various microbial species.

As the universal energy currency of all living cells and one of the most important metabolites, adenosine-5′-triphosphate (ATP) plays critical roles in nutrient transport, DNA replication, protein synthesis, chaperone-assisted protein folding, metabolite biosynthesis, stress response, and other microbial cellular processes such as flagellar movement. In microbial biotechnology, sufficient ATP supply is essential to maintain or optimize microbial activities at desirable levels for human applications, such as the production of recombinant proteins and many metabolite products[1,2].

During bacterial aerobic respiration, ATP is mainly regenerated through oxidative phosphorylation and glycolysis to ensure metabolite biosynthesis, cell growth, and other cellular activities[3]. The production and consumption of ATP can be affected by numerous factors, including nutrients, dissolved oxygen level, pH, temperature, growth stage, and cellular activities such as induced protein expression or metabolic activity[4–6]. Although ATP supply in microbial cells is naturally regulated, the imbalance between ATP production and consumption makes cellular ATP concentration highly dynamic during cell growth and environmental changes[2]. Even under steady environmental conditions, ATP levels in single microbial cells can display large fluctuations, partially depending on cell cycles and overflow metabolism[7]. This results in significant cell-to-cell variations in ATP concentration and potentially affects phenotypic heterogeneity[7–10]. Furthermore, ATP levels and fluctuations are strongly influenced by carbon sources. Cultivating *E. coli* in rich media, minimal glucose, or minimal glycerol have distinct cell cycle-related fluctuation patterns[7], suggesting that ATP dynamics are closely related to cell growth and metabolism. In microbial biotechnology, engineered microbial activities compete with native processes on ATP, further complicating ATP dynamics. For example, biosynthesis of 3-hydroxybutyrate via malonyl-CoA consumes lots of ATP in an engineered cyanobacterium strain, leading to metabolic imbalance and cell growth inhibition[11].

Given its critical importance in cell growth and biosynthesis, multiple strategies have been developed to elevate cellular ATP levels to improve bioproduction efficiency. Current approaches for enhancing ATP supplies include the use of acidic conditions for

[1]Department of Energy Environmental and Chemical Engineering, Washington University in St. Louis, Saint Louis, MO 63130, USA. [2]Division of Biological & Biomedical Sciences, Washington University in St. Louis, Saint Louis, MO 63130, USA. [3]Institute of Materials Science & Engineering, Washington University in St. Louis, Saint Louis, MO 63130, USA. ✉e-mail: fzhang@seas.wustl.edu

fermentation[12,13], the substitution of ATP-dependent nutrient transporters with ATP-independent transporters[14,15], and the replacement of metabolic pathways that do not produce ATP with those that do[15,16]. For example, improved succinate production has been observed in bacterial hosts through the introduction of an ATP-generating phosphoenolpyruvate (PEP) carboxy kinase from *Actinobacillus succinogenes* in place of the native, non-ATP-generating PEP carboxylase[1,17]. This substitution not only facilitates the conversion of PEP to oxaloacetate but also generates ATP. Similarly, protein synthesis in *E. coli* also benefits from an increased intracellular ATP concentration, which results from the overexpression of gluconeogenic enzyme phosphoenolpyruvate carboxykinase[18]. While promising, these strategies require specific conditions (e.g., acidic pH, optimal $O_2$ levels for the enzymatic activity of ATP-generating PEP-carboxy kinase, and nutrients whose transport is ATP-independent.), which may restrict their applicability in industrial fermentation settings. Furthermore, the dynamic nature of cellular ATP concentrations during bioproduction suffers insufficient investigation, partially due to the challenges in real-time quantification of ATP dynamics in living cells. Fortunately, the recent developments in genetically encoded ATP biosensors have made it possible to continuously monitor cellular ATP levels in microbial cells, offering insights into the regulation of ATP during bioproduction[19].

Here, we employ an $F_0$-$F_1$ ATP synthase-based ratiometric ATP biosensor[19] to study ATP dynamics across different growth phases using various carbon sources (e.g., glucose, glycerol, pyruvate, acetate, malate, succinate, and oleate) in multiple *Escherichia coli* strains and *Pseudomonas putida* KT2440. We find that the choice of carbon source significantly impacts ATP levels within each species. Notably, during exponential growth under aerobic conditions, *E. coli* and *P. putida* exhibit the highest ATP levels among all the tested carbon sources when cultivated with acetate and oleate, respectively. We further identify transient ATP accumulations during the transition from exponential to early stationary growth phases in multiple carbon sources for both species. We show that the transient ATP accumulation was caused by an imbalance between ATP production and consumption during growth transition and is responsible for the peak fatty acid (FA) productivity in an engineered *E. coli* strain. Moreover, we achieve enhanced bioproduction of FA in *E. coli* and polyhydroxyalkanoate (PHA) in *P. putida* by supplementing the carbon sources that improve ATP concentration in each strain. Additionally, we demonstrate the potential of ATP dynamics as a diagnostic tool for identifying the source of metabolic burden in limonene bioproduction. Our study reveals the intricate microbial ATP dynamics under various conditions and highlighted the use of ATP dynamics in enhancing bioproduction across multiple microbial systems.

## Results

### ATP dynamics across different growth phases in various carbon sources

To examine ATP dynamics during microbial growth, we utilized a genetically encoded ATP biosensor iATPsnFR1.1[19]. This ATP biosensor contains a circularly permuted super-folder green fluorescent protein (cp-sfGFP) integrated within the ATP-binding epsilon subunit of the $F_0$-$F_1$ ATP synthase. ATP binding at the $F_0$-$F_1$ ATP synthase domain induces a conformational change to the GFP domain, which leads to enhanced green fluorescence with a response time within 10 ms (Fig. 1A). To compensate for variations of sensor expression level under different conditions, we fused a red fluorescent protein, mCherry, to the sensor. The GFP to mCherry fluorescence ratio was calculated (Methods) and used to represent the ATP concentration in living microbial cells.

To investigate the effects of different carbon sources on ATP dynamics, we transformed an *E. coli* NCM3722 strain with the ATP biosensor and cultivated it in M9 minimal media supplemented with various carbon sources, including glucose, glycerol, pyruvate, acetate,

malate, succinate, and oleate. These carbon substrates were selected to represent critical entry points and junctures within the central carbon metabolism (Fig. 1B). In M9 glucose, ATP levels remained constant during the exponential growth phase. However, a notable shift was observed during the transition to the stationary phase: ATP levels initially surged at the onset of the transition, followed by a rapid decline upon entry into the stationary phase (Fig. 1C). To validate the ATP dynamics measured with the ATP biosensor, we quantified ATP concentrations at various growth stages using a commercial luciferase assay (Supplementary Fig. 1). The ATP measurement obtained from the luciferase assay matched closely with that from the ATP biosensor. This growth-phase-dependent ATP dynamics was particularly evident when ATP concentration changes were plotted against growth rates (Supplementary Fig. 2), revealing an increase in ATP levels coinciding with the onset of reduced growth rates and a significant decrease as the growth rate further declined.

We postulated that the transient ATP surplus was caused by production-consumption imbalances: the initial increase in ATP level results from a reduced demand for ATP due to slower cell growth during the early transition, while the subsequent decrease in ATP level is attributed to a reduced ATP supply as the culture reaches the stationary phase. To test this hypothesis, we adjusted glucose concentration to modulate ATP production time. While altering glucose concentration from 0.1% to 2% did not significantly affect cell growth rate during the early stationary phase, higher glucose concentrations are expected to extend the duration of ATP production, thus leading to a higher ATP peak (Supplementary Fig. 3a, b). To further prove this hypothesis, we manipulated the timing of cells entering the stationary phase by cultivating in a nitrogen-limited M9 glucose medium. Compared to nitrogen-repleted conditions, growth cessation occurred earlier in nitrogen-depleted conditions, leading to an earlier decline in ATP level and a lower ATP peak (Supplementary Fig. 3c, d). These results suggest that the ATP peaks are likely to be caused by a transient ATP surplus during the transition in growth phases. While intracellular ATP concentration can also be potentially affected by cell volume change, we found that the average cell volume change from the last round of cell division to the stationary phase is much smaller (<10%, Supplementary Fig. 4, Supplementary Movie 1) than the changes observed in ATP concentration, thus is unlikely to be the major factor affecting ATP concentration. Across all conditions, cells in the late stationary phase consistently exhibited lower ATP levels than those in the exponential growth phase, which aligns with prior observations[20–23].

When growing *E coli* NCM3722 strain in other carbon sources, we observed similar dynamics−an ATP peak appearing during the transition from exponential growth to early stationary phase (Fig. 1D−I). However, the size of ATP peaks varied with carbon sources. We found a strong positive correlation ($r^2 = 0.89$, $p < 0.001$) between the ATP peak size and steady-state cell growth rate among different carbon sources (Fig. 1J), indicating that fast-growing cells experience a more substantial ATP change during growth transition. This correlation also supports the above-described transient ATP surplus hypothesis because fast-growing cells undergo a more significant change in their growth rates during growth transition than slow-growing cells. Thus, the abrupt decrease in ATP demand from growth creates more excess ATP in fast-growing cells.

During the exponential growth phase, intracellular ATP concentrations stabilized at relatively steady levels across various carbon sources, albeit at different levels. There is little correlation between exponential growth phase ATP levels and growth rates ($r^2 < 0.01$, $p = 0.97$, Fig. 1K), consistent with previous findings that ATP concentration does not limit cell growth[24,25]. The different exponential growth phase ATP level also suggests that cells modulate ATP production rates in response to the available carbon source[13,20]. Surprisingly, cells grown in acetate exhibited a higher ATP level during the

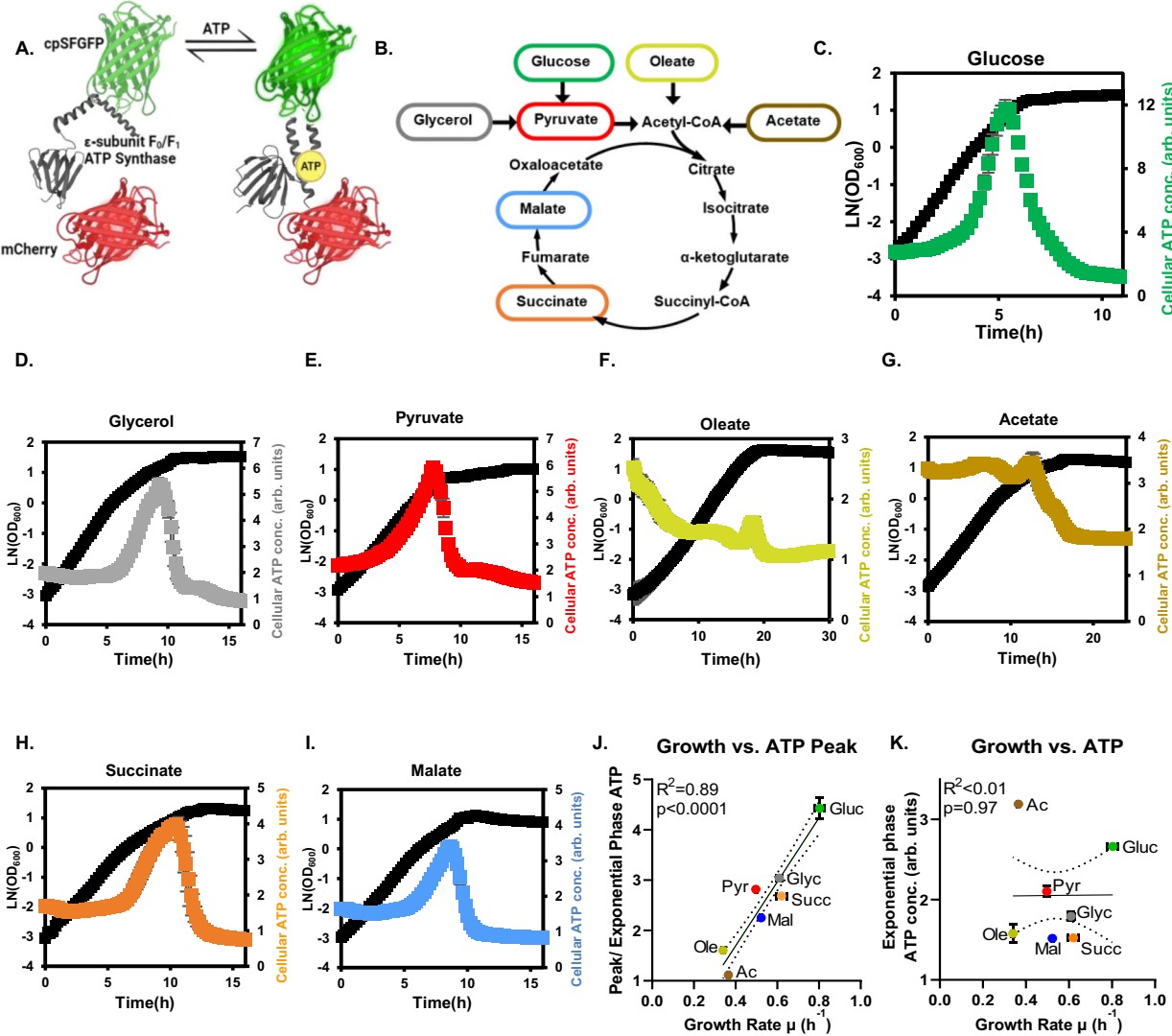

**Fig. 1 | ATP dynamics of *E. coil* under different carbon sources. A** Mechanism of ATP biosensor. **B** Various carbon sources used in this study. **C–I** Cell growth and ATP dynamics of *E. coli* NCM3722 cultivated in glucose, glycerol, pyruvate, oleate, acetate, succinate, and malate. Error bars represent the standard deviation from 3 biological replicates (*n* = 3). **J** Cell growth rate vs. the height of ATP peak in the early stationary phase. ATP peak height was calculated as the ratio of peak ATP level over steady-state ATP level. *P*-value (*p* = 0.000086) is calculated from F-test. **K** Cell growth rate vs. steady-state ATP level. All experiments were performed in biological triplicates. Source data are provided as a Source Data file.

exponential growth phase compared to those grown in glucose (Fig. 1K), which is contrary to the intuitive understanding that aerobic glucose metabolism has a higher molar yield of ATP than acetate metabolism per mole of carbon source. To understand the mechanisms underlying the steady-state ATP concentration ([ATP]$_{ss}$) across different carbon sources, we developed a kinetic model considering the balance between ATP production ($r_{production}$) and consumption ($r_{consumption}$) rates (Supplementary Note). Our model analysis suggested that the higher [ATP]$_{ss}$ in acetate than glucose resulted from the elevated ATP production rate from acetate metabolism in *E. coli*, which was supported by previous flux analyses under similar conditions[26,27] (Supplementary Note).

Additionally, we extended our analysis of ATP dynamics to include two additional *E. coli* strains, MG1655 and DH1. Both strains exhibited ATP dynamics similar to NCM3722, albeit with smaller ATP peaks during the growth transition due to their relatively slower steady-state growth rates (Supplementary Fig. 5a–g, 6a–g). All three tested *E. coli* strains exhibited consistent patterns in ATP levels among different carbon sources, with acetate cultures consistently showing the highest

ATP levels (Supplementary Fig. 5h, 6h). Traditionally, acetate has been viewed as a less favorable carbon source, often associated with overflow metabolism. Nevertheless, recent studies have begun to recognize the beneficial role of acetate in facilitating robust growth during glycolytic perturbation[1,28]. Our ATP dynamics results corroborate that acetate promotes cellular ATP concentrations, suggesting its potential to enhance the biosynthesis of energy-demanding products.

## FA bioproduction benefits from acetate supplementation

FAs are essential precursors in the biosynthesis of biofuels, detergents, lubricants, and polymer precursors[29,30]. FA biosynthesis demands substantial energy input (Fig. 2a), consuming 7 ATP and 14 NADPH molecules to synthesize one C16 FA molecule. We hypothesized that FA biosynthesis could be limited by the availability of intracellular ATP. Indeed, when measuring FA production from glucose across the entire growth stage, we observed a peak in FA productivity at the early stationary phase (Fig. 2b), coinciding with the ATP peak in glucose culture.

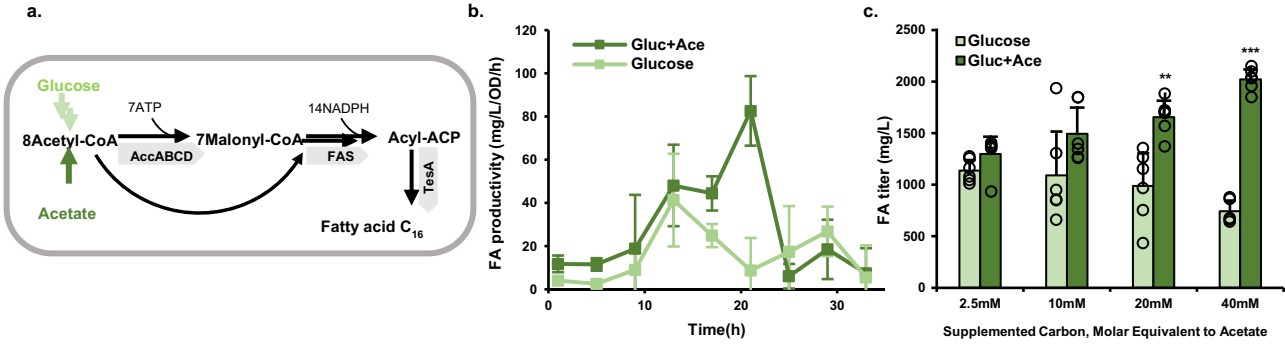

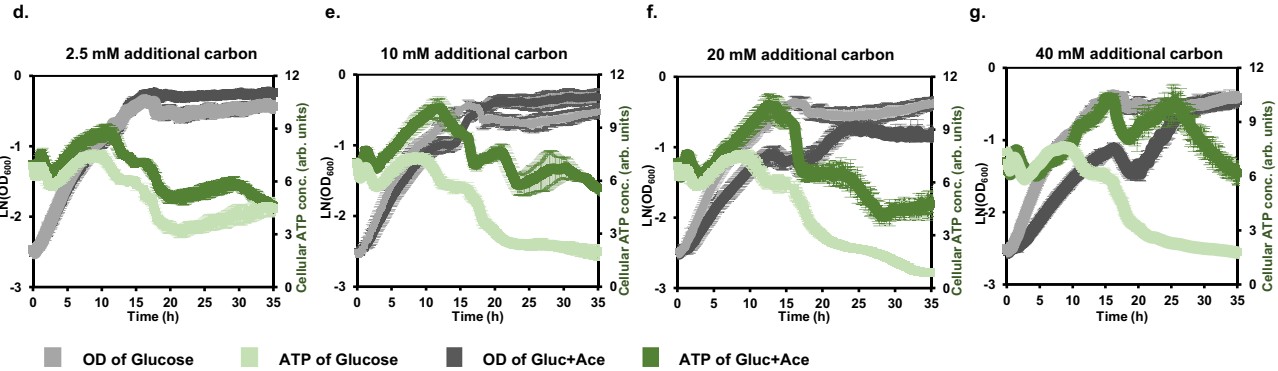

**OD of Glucose**  **ATP of Glucose**  **OD of Gluc+Ace**  **ATP of Gluc+Ace**

**Fig. 2 | ATP dynamics and FA production in *E. coli*. a** The FA biosynthesis pathway. **b** FA productivity in minimal glucose (light green) or mixtures of glucose and acetate (40 mM) of the same carbon equivalent (dark green). Error bars represent the standard deviation from three biological replicates ($n = 3$). **c** FA titers after 48 h of fermentation in minimal glucose supplemented with different amounts of acetate. ** indicates $p \leq 0.01$ ($p = 0.0023$), and *** indicates $p \leq 0.001$ ($p = 0.00001914$) from two-tailed t-test. Error bars represent standard deviation ($n = 6$). **d–g** ATP dynamics and cell growth of FA-producing *E. coli* cultures supplemented with different amounts of acetate or glucose. Error bars represent the standard deviation from three biological replicates ($n = 3$). Source data are provided as a Source Data file.

Next, we attempted to leverage both the high steady-state ATP concentration from acetate and the ATP peak from glucose during the early stationary phase. ATP dynamics was measured in wildtype *E. coli* NCM3722 cells cultured in minimal glucose media supplemented with various amounts of acetate. Increasing acetate concentration from 2.5 to 40 mM resulted in elevated ATP levels during the exponential growth phase, while the ATP peak at the early stationary phase and cell growth rates remained unchanged (Supplementary Fig. 7). This observation is consistent with recent findings that *E. coli* can metabolize glucose and acetate concurrently[31,32]. To further examine whether the elevated ATP levels can benefit bioproduction, we cultivated an FA-producing strain (sXM01, Supplementary Table 1) in media containing glucose-acetate mixtures and compared it to those containing only glucose with equivalent carbon molar concentrations. Cells grown in mixed carbon sources showed higher ATP levels than those in glucose alone across varying carbon concentrations, and increasing the acetate concentration from 2.5 mM to 40 mM led to higher intracellular ATP levels (Fig. 2d–g). With a 40 mM acetate supplement, ATP levels remained high throughout both the exponential and stationary growth phases, whereas ATP levels in glucose media declined rapidly after the transition to the stationary phase. The distinct ATP dynamic patterns between glucose and mixed carbon sources translated into different FA production outcomes, with mixed carbon sources yielding higher FA titers than glucose cultures at all tested carbon concentrations. Notably, with an additional 40 mM acetate, the FA titer and yield were 2.7-fold higher than that supplemented with equivalent glucose (Fig. 2c, Supplementary Fig. 8a). Moreover, we measured FA productivity every 4 h during fermentation in cultures supplemented with 40 mM acetate or equivalent glucose

(Fig. 2b, Supplementary Fig. 8b, c). FA productivity patterns mirrored the ATP dynamics, displaying a high FA productivity peak during the growth phase transition and maintaining elevated FA productivity over a prolonged period due to sustained high ATP levels under these conditions. Overall, these results demonstrated the efficacy of acetate supplementation in improving cellular ATP concentration and FA production in *E. coli*.

## ATP dynamics identified oleate as a beneficial carbon source promoting PHA production in *P. putida*

To broaden the applicability of our strategy to other species and metabolites, we next focused on *P. putida*, an industrial-relevant microbial strain widely used for the bioproduction of PHA, polyketides, and aromatic compounds[33]. The ATP dynamics of *P. putida* in various carbon sources, particularly its influence on metabolite production, remain largely unexplored. We modified the ATP biosensor (pB6k-mCherry-iATPsnFR1.1) for use in *P. putida* KT2440 and measured ATP dynamics in different carbon sources (Supplementary Fig. 9a–f). Due to the rapid consumption of nutrients, *P. putida* cultured exhibited very short steady-state growth periods under our experimental conditions. The rapid nutrient consumption led to continuous ATP decreases without a pronounced peak through the growth phase transition. Among the tested carbon sources, oleate promoted the fastest growth rate and the highest ATP level (Fig. 3a).

We hypothesized that the elevated ATP level in the oleate medium could be attributed to PHA biosynthesis. *P. putida* synthesizes PHA via either the de novo pathway using acetyl-CoA as the precursor or the β-oxidation pathway using acyl-CoA as the precursor (Fig. 3b). The β-oxidation pathway generates NADH, which can be converted to ATP

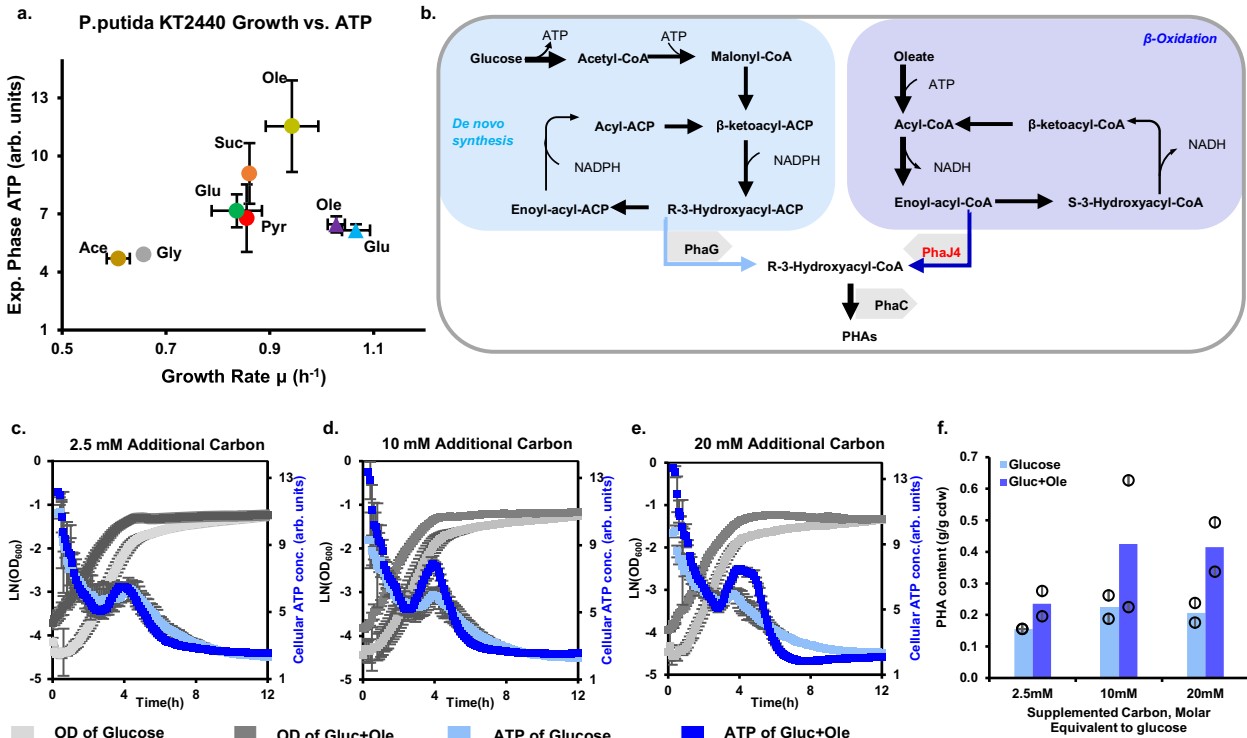

**Fig. 3 | ATP dynamics and PHA content of *P. putida*. a** Growth rates and exponential growth phase ATP levels of the wildtype (circles) and ΔphaJ4 (square) *P. putida* strain. Error bars represent standard deviation ($n = 3$). **b** PHA biosynthesis pathways. **c**–**e** ATP dynamics and OD of PHA-producing *P. putida* cultures supplemented with different amounts of glucose or oleate. Error bars represent standard deviation ($n = 3$). **f** PHA content after 24 h fermentation in additional amounts of glucose or oleate supplementation. Error bars represent standard deviation ($n = 2$). Source data are provided as a Source Data file.

through oxidative phosphorylation under aerobic conditions, suggesting that PHA biosynthesis serves as a carbon sink, driving the flux through the β-oxidation and producing ATP. To test this hypothesis, we deleted *phaJ4*, whose enzyme product converts enoyl-acyl-CoA to R-3-hydroxyacyl-CoA, thus blocking PHA biosynthesis directly from the β-oxidation pathway and lowering β-oxidation flux demand. Indeed, the ΔphaJ4 strain displayed significantly reduced ATP levels than the wildtype strain when growing in oleate (Supplementary Fig. 9g, h), supporting the proposed link between PHA biosynthesis and ATP production. Further, this decrease in ATP level is related to β-oxidation rather than other pathways or global effects because phaJ4 deletion did not alter ATP level in glucose, whose metabolism does not involve β-oxidation[34] (Fig. 3a, Supplementary Fig. 9g, h). Further supporting this hypothesis, PHA contents in ΔphaJ4 cultures did not change with increasing oleate concentrations (Supplementary Fig. 10a). Moreover, the ΔphaJ4 strain exhibited faster growth in oleate medium than the wildtype strain (Supplementary Fig. 9h), confirming that ATP concentration does not limit cell growth and indicating that the surplus ATP in wildtype *P. putida* might be redirected towards bioproduction when metabolizing oleate. Additionally, PHA profile also changed in ΔphaJ4 strain due to the invalid β-oxidation pathway, which mostly consisted of hydroxydecanoate (C10) and a little hydroxyoctanoate (C8), while wildtype *P. putida* could produce more C8 and C12 PHA (Supplementary Fig. 11)[35].

Building on the insights from ATP dynamics in *P. putida*, we explored the potential of oleate supplementation to enhance PHA bioproduction in *P. putida*. Oleate supplements significantly increased cellular ATP levels and PHA accumulations compared to glucose cultures with the same carbon equivalent. Moreover, increasing the concentration of supplemented oleate from 2.5 mM to 10 mM resulted in a pronounced ATP peak during the growth phase transition (Fig. 3c–e), leading to a substantial enhancement in PHA content, titer,

and productivity (Fig. 3f, Supplementary Fig. 10b, c). These results underscore the utility of ATP dynamics not only for selecting beneficial carbon sources for *P. putida* but also for optimizing PHA production in engineered *P. putida* strains.

## ATP dynamics monitoring as a diagnostic tool to visualize metabolic burden

The expression of engineered biosynthetic pathways in host cells is well known to impose various burdens[36,37], including competition for free ribosomes in protein synthesis[24,38], competition with native pathways for precursors or cofactors[39,40], energy depletion through ATP competition, unnecessary enzyme production[41–43], accumulation of metabolic intermediates to toxic levels[39,44], and multifactorial stresses[38]. Traditionally, the identification of these burdens has relied on direct measures of cell growth rates or reporter protein, which may not accurately reflect the energetic burden prevalent during bioproduction[24,45]. Both enzyme overexpression and engineered metabolic reactions can consume a large amount of ATP, potentially disrupting ATP dynamics and constraining bioproduction capacities.

In this context, we aimed to use ATP dynamics to elucidate the energetic burden imposed by an engineered limonene-producing pathway, which, despite being implemented in various microbial hosts, has consistently yielded low limonene titers[46–48]. The pathway involves the expression of 8 heterologous enzymes, cumulatively requiring 9 ATP molecules to synthesize a single limonene molecule (Fig. 4a)[49]. Using the ATP biosensor, we found that induction of the limonene pathway in *E. coli* caused substantial decreases in cellular ATP levels, indicative of significant energy burdens (Fig. 4b). Furthermore, we noticed that both the limonene titer[49] and the averaged ATP level decreased as the induction level increased (Fig. 4c, Supplementary Fig. 12a), underscoring ATP availability as a limiting factor for

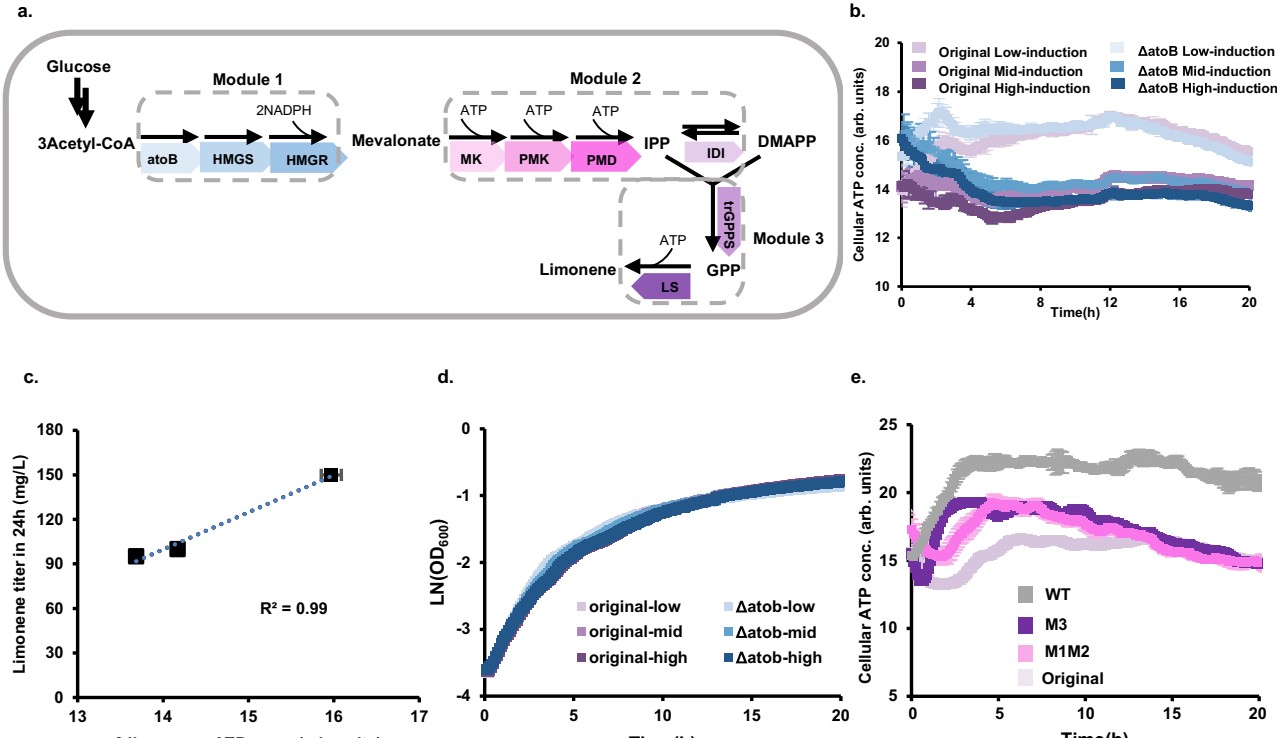

**Fig. 4 | ATP dynamics of limonene-producing *E. coli*. a** Limonene biosynthesis pathway. **b** ATP dynamics of the original and *atoB*-deleted pathways under 25, 100, and 500 μM IPTG induction. Error bars represent standard deviation ($n = 3$). **c** Correlation of averaged ATP levels and limonene titers after 24 h of fermentation under different induction levels. Limonene titers were estimated from Supplementary Fig. 12a. Error bars represent standard deviation ($n = 3$). **d** Cell growth of *E. coli* strains expressing the original or *atoB*-deleted limonene pathway under 25,

100, and 500 μM IPTG induction. Error bars represent standard deviation ($n = 3$). **e** ATP dynamics of *E. coli* strains expressing the original and variants limonene pathways. M3: expressing only Module 3; M1M2: expressing both Module 1 and 2. WT: wildtype *E. coli* only containing the ATP biosensor without the limonene pathway. Error bars represent standard deviation ($n = 3$). All experiments were performed in biological triplicates. Source data are provided as a Source Data file.

limonene production. To differentiate whether the observed energy burden was caused by enzyme expression or metabolic reactions, we inactivated the limonene pathway by deleting the first gene *atoB*. The *atoB* deleted strain exhibited ATP dynamics similar to those of the intact pathway across a range of induction levels, suggesting that the burden was primarily exerted by enzyme expression rather than metabolic activities (Fig. 4b). Surprisingly, both strains under different induction levels showed very similar growth curves (Fig. 4d), indicating that the traditionally used cell growth is a poor indicator of metabolic burden[24,45]. In this case, expression of the limonene pathway caused a notable decrease in cellular ATP levels without affecting growth.

To further pinpoint the source of energy burden, we constructed partial pathways containing only one or two modules of the entire limonene pathway. Each partial pathway exhibited some degree of energy depletion compared to the wildtype *E. coli*, yet their growth curves did not differ from the wildtype strain (Fig. 4e, Supplementary Fig. 12b), which further confirmed the advantage of using ATP dynamics to diagnose metabolic burdens. Notably, a strain expressing the third module, consisting of only the truncated geranyl diphosphate synthase (trGPPS) and a limonene synthase (LS), showed similar ATP depletion to that expressing the first two modules (6 enzymes in the isoprenoid pathway, Fig. 4e). These results suggest that expression of the limonene pathway, particularly the third module, needs to be optimized to improve host cell energetics and limonene production. Furthermore, ATP dynamics provides a more sensitive diagnostic tool for identifying pathway bottlenecks, presenting a valuable strategy for metabolic engineering.

## Discussion

In this work, we used an ATP biosensor to examine ATP dynamics in multiple microbial strains and under various nutrient conditions. Our results highlight the benefits of monitoring ATP dynamics for investigating microbial energy homeostasis, selecting optimal carbon sources for enhanced bioproduction, and identifying the source of the burden imposed by engineered metabolic pathways. Traditional approaches for studying microbial energy homeostasis include ATP quantification through liquid chromatography-mass spectrometry (LC-MS), luciferase assays, or metabolic flux analyses[23,50–52]. These early attempts revealed ATP concentration changes during growth phase transitions in a few *E. coli* strains[22,23]. However, conflicting results were obtained when exploring the relationship between steady-state ATP concentration and *E. coli* growth rate. It was found that different ATP extraction methods used in traditional quantification led to different ATP concentrations and opposite conclusions[53,54]. Additionally, traditional extraction-based ATP quantification requires multiple treatment steps and is time-consuming, which limits the temporal resolution of ATP dynamics. In contrast, the ATP biosensor facilitates straightforward monitoring of ATP concentration changes in living microbial cells without extraction. The detailed growth-phase-dependent ATP dynamics in various carbon sources are challenging to elucidate through other methods.

This work also revealed that *E. coli* cells growing in acetate have a higher intracellular ATP concentration than those growing in glucose, a surprising observation given the traditional thinking of higher ATP yield (per mole of carbon) from glucose metabolism. The high ATP level in acetate is attributed to a combination of elevated ATP

production rate from acetate metabolism and reduced ATP consumption rate due to slower cell growth in *E. coli* (Supplementary Note). During the exponential growth phase, the ATP production coefficient from acetate was found to be at least 2-fold higher than that from glucose, as determined by either previous experimental measurements or flux balance analysis[27,55] (Supplementary Note). The enhanced ATP production rate in acetate culture primarily results from both a high carbon update rate and a high flux through the TCA cycle, facilitating reducing cofactor generation that contributes to ATP production via oxidative phosphorylation. Thus, acetate's unique metabolic characteristics present an opportunity for boosting bioproduction through supplementation with acetate, a low-cost and renewable carbon stock that can be obtained from anaerobic digestion or electrocatalytic conversion of $CO_2$ or syngas[56,57]. Likewise, oleate and potentially other medium-chain fatty acids are beneficial carbon sources for *P. putida*, capable of enhancing cellular ATP concentration and PHA production. These results emphasize the variability of microbial ATP dynamics with species and carbon sources, necessitating a tailored approach for each case. The strategy of using ATP dynamics to select beneficial carbon sources can be extended to a wide range of microorganisms, improving their production of chemicals, proteins, and materials.

Additionally, this study demonstrates that monitoring ATP dynamics offers a more sensitive metric for assessing metabolic burden than traditional methods based on cell growth. In the production of limonene by *E. coli*, heterologous enzyme expression significantly reduced ATP levels, impacting limonene yields without substantially affecting cell growth. Similar energy burdens are likely to present in the production of other chemicals from ATP-demanding pathways. Consequently, heterologous protein expression must be carefully tuned to maintain ATP balance, with the ATP sensor serving as a powerful tool for identifying proteins and reactions that impose energy burdens on host cells. The metabolic burden caused by other factors, such as the depletion of pathway precursors or cofactors, may not impact cellular ATP concentration. Further investigation is needed to determine whether ATP dynamics can effectively indicate metabolic burden from these sources.

## Methods

### Plasmids, strains, and culture conditions

All plasmids were constructed using BglBrick[58] or Golden Gate assembly[59] methods following well-established protocols. Strains *E. coli.* strains NCM3722, MG1655, and DH1 and *P. putida* KT2440 were engineered and used where indicated. For cell cultures, single colonies were used to inoculate 5 ml of LB medium containing appropriate antibiotics (50 mgl⁻¹ ampicillin, 50 mgl⁻¹ kanamycin, 30 mgl⁻¹ chloramphenicol, 35 mgl⁻¹ gentamycin, 25 mgl⁻¹ tetracycline) and incubated at 37 °C for *E. coli* or 28 °C for *P. putida* with orbital shaking at 250 rpm. Overnight LB cultures were used to inoculate different media described in each section. Briefly, experiments were performed either in a 50% EZ-Rich defined medium (Teknova, Hollister, CA), or in a modified M9 Media containing M9 salts (Sigma-Aldrich, St. Louis, MO), 75 mM MOPS (pH 7.4), 2 mM MgSO₄, 0.1 mM CaCl₂, 1 mgl⁻¹ thiamine, 74.7 mM NH₄Cl, 10 µM FeSO₄ and micro-nutrients (3 µM (NH₄)₆Mo₇O₂₄·4H₂O, 400 µM boric acid, 30 µM CoCl₂·6H2O, 15 µM CuSO₄, 80 µM MnCl₂·4H₂O and 10 µM ZnSO₄·7H₂O), which were supplemented with different carbon sources and proper antibiotics. Cultures were induced with a final concentration of 25–1000 µM isopropyl β-d-1-thiogalactopyranoside (IPTG) or 25 mM L-arabinose as described in the manuscript. The *phaJ4* gene was deleted from *P. putida* KT2440 chromosome using a modified CRISPR/Cas9 system as described previously[60]. Briefly, *P. putida* KT2440 containing pCas9 was transformed with pJOE_phaJ4 (Supplementary Table 2) by electroporation and selected on LB agar plates containing 35 µg/ml Gentamycin (Gen35) and 50 µg/ml Kanamycin (Kan50). One transformant was

grown in LB Gent35, Kan50 at 30 °C. After reaching the stationary phase, λRed genes were induced with 0.5% L-arabinose for 15 min. This culture was used to make electrocompetent cells by washing twice with 10% glycerol and resuspending in 100 µL 10% glycerol per 1 mL of culture. The cells were then transformed with pgRNAtet_phaJ4 (Supplementary Table 2) by electroporation and recovered in 1 mL LB for 2 h at 30 °C. Recovered cells were selected on LB agar plates containing 25 µg/ml tetracycline (Tet25) and Gen35. Transformants were screened for the desired knockout using colony PCR with primers flanking *phaJ4*. Positive hits were re-streaked and screened again with a secondary colony PCR for the presence of wild type. Plasmids were cured by growing overnight in LB without antibiotics and plating on LB agar. Single colonies were screened for loss of antibiotic resistance. The single guide RNA (sgRNA) used for *phaJ4* deletion has the following targeting sequence: 5′-agctctgtaaccggtacatg-3′.

### ATP dynamics in different carbons

ATP dynamics in different carbon sources were measured using *E. coli* strains NCM3722, MG1655, and DH1 carrying the ATP biosensor plasmid pS6k-mCherry-iATPsnFR1.1 (Supplementary Table 1) and *P. putida* KT2440 carrying the ATP sensor plasmid pB2k-mCherry-iATPsnFR1.1. Carbon sources, including acetate, pyruvate, glycerol, succinate, malate, and oleate were used at carbon molar concentrations equivalent to 0.4 wt% glucose. Initially, an overnight LB culture was inoculated into M9 media containing 0.4% glucose for 8 h. Subsequently, the M9-glucose culture was inoculated into M9 media containing the aforementioned carbon sources, except for oleate cultures, which were adapted from exponentially growing M9-glycerol culture. All cultures were induced with 500 µM IPTG to initiate ATP sensor expression and were maintained in the exponential growth phase for a minimum of 12 h by continuous dilution to ensure that OD₆₀₀ were consistently below 0.4. These exponentially growing cultures were then transferred to a fluorometric plate reader (TECAN Infinite 200PRO) for quantifying cell culture fluorescence. The excitation and emission wavelengths for mCherry fluorescence were set at 535 ± 9 nm and 620 ± 20 nm, while those for GFP fluorescence were 488 ± 9 nm and 518 ± 20 nm, and for CFP were 430 ± 9 nm and 470 ± 20 nm. The ATP level was calculated as the ratio of GFP fluorescence and mCherry fluorescence, or the ratio of GFP fluorescence and CFP fluorescence after subtracting the background fluorescence of the media. All experiments were performed in biological triplicates.

### ATP extraction and luciferase assay

ATP extraction and quantification were performed following published protocols[61]. Briefly, at each time point, a 150 µL cell culture sample was taken and subjected to the treatment of 100 µL of 1.2 M cold perchloric acid. The mixture was briefly vortexed and incubated on ice for 15 min before centrifugation at 2500 × g for 10 min. A 200 µL supernatant was taken and mixed immediately with 100 µL cold neutralization buffer containing 0.72 M potassium hydroxide and 0.16 M potassium bicarbonate. ATP concentrations of the extracted samples were then quantified using an ATP Determination Kit (Thermo Scientific, cat# A22066) following the manufacturer's instructions. The measured luminescence signals were compared to a standard curve generated using a 5 mM standard ATP solution provided in the kit. ATP concentrations measured from the luciferase assay ([ATP]$_{luc}$) were then converted to cellular ATP concentrations ([ATP]$_{cell}$) using the following equation:

$$[ATP]_{cell} = \frac{2.5[ATP]_{luc} \times V_{sample}}{n \times V_{sample} \times OD_{600} \times V_{cell}},$$

where $n = 1.1 \times 10^9$ /mL/OD, representing the number of cells per mL culture per OD; $V_{cell} = 1.0 \times 10^{-15}$ L, representing the volume of a single *E. coli* cell; and $V_{sample}$ represents the volume of each sample used for

ATP quantification[62]. Thus, $n \times V_{sample} \times OD_{600} \times V_{cell}$ represent the total intracellular volume from all sampled cells. The constant 2.5 is the dilution factor.

## Timelapse microscopy for cell size measurement

To measure cell size changes, *E. coli* cells were grown on an agarose pad, and a time-lapse microscope video was acquired using published methods[63]. Briefly, low-melt agarose (EMD OmniPur low-melting agarose, EMD cat. no. 2070) was added to 1 ml of M9 0.1% glucose medium to a final concentration of 1.5% (wt/vol). The agarose was dissolved by heating to 90 ~ 100 °C and vertexing. After partial cooling, 150 μL of the warm agarose medium was transferred onto a 22 mm² cover glass slide. Another cover glass was placed on top of the agarose to create a flat surface. After the agarose pad was completely solidified and cooled, 1 μL of exponentially growing *E. coli* cell culture was loaded onto the agarose pad. The seeded agarose pad was then placed into a humid incubation chamber (Tokai Hit, Incubation Systems for Microscopes, Japan) at 37 °C. Phase contrast images were collected using a Nikon Ti-eclipse Microscope (Melville, NY, U.S.A.) equipped with the Nikon Perfect Focus (PFS) unit and an oil immersion 100x objective (Nikon). Phase contrast images were collected every 5 min for 14 h. Cell areas were measured using the NIS-Element 4500 (Nikon). The change of cell area across multiple cell lineages was plotted over time, as shown in Supplementary Fig. 4. The average cell area change from the last cell division to the stationary phase is less than 10%.

## FA production and quantification

FA fermentation was performed using an *E. coli* DH1 derivative strain sXM01, which contains a *fadE* deletion, a cytosolic *tesA* gene under the control of an inducible promoter, and an ATP sensor plasmid pSJ23119a-CFP-iATPsnFR1.1[64]. For FA production, an overnight M9-glucose culture of this strain was diluted to a fresh M9 medium containing 1 wt% glucose with a starting $OD_{600}$ of 0.1 and cultivated at 37 °C with shaking. When the culture reached an $OD_{600}$ of 0.6, IPTG was added to a final concentration of 1000 μM to induce FA production. Meanwhile, additional glucose or acetate was supplemented to final concentrations as described in the paper. To measure ATP dynamics, 150 μL of the induced culture was transferred to a 96-well plate and cultivated inside a plate reader at 37 °C with constant shaking for 48 h. FA titers were quantified following published methods[44]. Specifically, 0.5 ml of cell culture was acidified with 50 μL of 6 N HCl and extracted twice by 0.5 ml of ethyl acetate spiked with nonadecanoic acid (C19:0) as an internal standard. The organic layer was transferred to a glass gas chromatography (GC) vial (Agilent). A solution containing 10 μL of 6 N HCl and 90 μL of methanol was added to the vial followed by adding 100 μL of trimethylsilyl-diazomethane (2 M in hexanes, Sigma) for FA methylation. The samples were incubated at an ambient temperature for 15 min and then analyzed by GC-FID (Hewlett Packard model 7890 A, Agilent) equipped with a DB5-MS column (Agilent J&W). For each sample, the column was equilibrated at 80 °C for 1 min, followed by a ramp to 280 °C at 30 °C /min and then held at this temperature for 3 min. The final concentration of fatty acid methyl ester (FAME) was analyzed on the basis of the internal standard and standard FAME mix (GLC-20 & GLC-30, Sigma-Aldrich). Experiments were performed in biological triplicates.

## PHA production and quantification

PHA production and ATP dynamics studies were performed using *P. putida* KT2440 strain sXM06, which overexpresses phaC1 and phaJ4 and contains an ATP sensor plasmid pB2k-mCherry-iATPsnFR1.1. An LB overnight culture of this strain was diluted 1:100 into 50% EZ-Rich defined media containing 2% glucose and cultured for 24 h. PHA production was induced at $OD_{600}$ of 0.6 with 500 μM IPTG. Meanwhile, cell cultures were supplemented with additional glucose or oleate at indicated concentrations. Cells were subsequently cultivated at 28 °C

for 24 h. The analysis of cell dry weight and PHA content was performed following published methods[33]. Briefly, cells were harvested by centrifugation at 13,800 × g for 10 min (Thermo Scientific Multifuge X1R Refrigerated Centrifuge) and washed twice using 0.9% NaCl solution. The cell pellet was freeze-dried using a lyophilizer for 24 h (Labconco Corporation, Kansas City, MO), and cell dry weight was measured using a balance (Mettler Toledo Standard ME Analytical Lab balance). To quantify PHA content, 5–10 mg of freeze-dried cells was added to a solution containing 2 ml of 15% (v/v) sulfuric acid dissolved in methanol and 2 ml of chloroform. The mixture was then incubated at 100 °C for 140 min. After cooling, 1 ml of distilled water was added, and the mixture was then centrifuged at 1200× g for 10 min to separate the organic phase from aqueous phase. A volume of 0.5 ml organic layer was then transferred and mixed with 0.5 ml of 0.1% caprylic acid in chloroform. PHA content was then analyzed using GC–MS (7820 A and 5977MSD, Agilent.) equipped with a DB5-MS column (Agilent J&W) (30 m × 250 μm × 0.25 μm). For each sample, the column was equilibrated at 50 °C for 3 min, followed by a ramp to 300 °C at 10 °C /min. The final concentration of PHA was analyzed on the basis of the external standard caprylic acid. Experiments were performed in biological duplicates.

## Quantifying ATP dynamics of *E. coli* strains containing various limonene pathways

*E. coli* DH1 strains carrying various limonene pathways were transformed with the ATP sensor plasmid pSJ23119a-CFP-iATPsnFR1.1(Supplementary Table 1). To measure the ATP dynamics of these strains, overnight cultures in M9-glucose medium were used to inoculate fresh M9 medium containing 2% glucose with a starting $OD_{600}$ of 0.1. Limonene production was induced at $OD_{600}$ of 0.6 by adding IPTG to indicated concentrations. A volume of 150 μL induced culture was subsequently transferred to a 96-well plate and incubated inside a plate reader at 37 °C with shaking. Real-time mCherry and GFP fluorescence were recorded to calculate relative ATP concentrations. All experiments were performed in biological triplicates.

## Statistics and reproducibility

Statistical analyses, including T-test and F-test, were conducted with Microsoft Excel. All data are presented as means ± SD with the number of replicates indicated. Significant statistical difference is considered at $p \le 0.01$ and is denoted with asterisks (**), while no significant statistical difference is considered at $p \ge 0.05$. Details were described in each figure legend. No data were excluded from the analyses.

## Reporting summary

Further information on research design is available in the Nature Portfolio Reporting Summary linked to this article.

## Data availability

All data generated in this study are provided in the article, the Supplementary Information, and the source data. Source data are provided with this paper.

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

## Acknowledgements

This work was supported by the National Institute of General Medical Sciences of the National Institutes of Health under Award Number R35GM133797 (F.Z.) and the Bioenergy Technologies Office (BETO) of U.S. Department of Energy under Award Number DE-EE0010301 (F.Z.).

## Author contributions

X.M., T.D.E., and F.Z. conceived the project, designed the experiments, analyzed the data, and wrote the manuscript. X.M. performed experiment for ATP dynamics of *E. coli* strains MG1655 and DH1, *P. putida* KT2440 wildtype and ΔphaJ4 strains using different carbon sources; ATP dynamics of FA and PHA producing strains, and strains containing various limonene pathways; FA and PHA production and quantification. T.D.E. performed experiment for ATP dynamics of *E. coil* strain NCM3722 using different carbon sources and under nitrogen depletion/repletion conditions.

## Competing interests

The authors declare no competing interests.
