## [Peer Review File · Nature Communications]

REVIEWER COMMENTS

Reviewer #1 (Remarks to the Author):

Mu et al. use a previously developed ATP ratiometric sensor to quantify the ATP concentrations in *Escherichia coli* and *Pseudomonas putida* under different carbon conditions during different growth phases. Using the sensor, the authors corroborate previous findings that, in *E. coli*, acetate results in higher ATP levels than glucose. The authors find that in *P. putida*, oleate results in higher ATP levels.

The key engineering finding of this work is the demonstration that spiking acetate, in the case of *E. coli*, or oleate, in the case of *P. putida*, increases ATP levels resulting in higher fatty acid or PHA titers, respectively.

The key scientific finding of this manuscript is the discovery that ATP levels is a better metric than cell growth to determine reaction burden. This is shown when the authors discover that Module 3 of the limonene synthesis pathway shows similar ATP depletion levels than Modules 1 and 2 combined, even though Modules 1 and 2 use require 3 ATPs while Module 1 requires 1. The authors conclude by highlighting the potential of the ATP sensor for identifying bottlenecks in metabolic engineering.

This work will be of great significance to the metabolic engineering community as it shows that supplementation with often overlooked carbon sources can result in significantly higher titers. It would be interesting to determine if the other biosynthetic productivity metrics (yield, rates) also increase when increasing ATP levels. The work supports the conclusions in the manuscript. The methodology is sound and there is enough details in the methods for the work to be reproduced.

Reviewer #2 (Remarks to the Author):

In this study, the authors measured cell growth and ATP level change with fluorescent biosensor iATPSnFR. The ATP level dynamics were measured in bulk by plate readers and growth dynamics were simultaneously measured by optical density.

The author reported several interesting findings about cellular ATP in *E. coli*, including (1) ATP level transiently increases during entry of stationary phase, (2) ATP level is higher in acetate media, and in glucose medium the amount of acetate supplement is positively correlated with fatty acid synthesis (3) ATP level is reduced in strains that synthesizing limonene. In addition, the author found in *P. putida* the oleate supplement increases cellular ATP level. Overall, the author proposed to use ATP level as an indicator of metabolic burden, which can be examined for bioengineered strains that synthesize various kinds of compounds.

Following are my comments:

1. There is one important technical issue in this work. The ATP concentration of *E. coli* measured by luciferase in this study (Fig. S1) is 10 to 100 times lower than the values reported in literatures, which is about several mM, or several micromolar per gram of cell dry weight [ref1, ref2, ref 3].

Related to this, the formula in line 445 for calculating [ATP] is confusing. The unit is incorrect, and I am wondering the term V_{sample} in the denominator should be a typo since the numerator already has [ATP] as concentration.

The calibration of ATP concentrations will affect the interpretations of some results in this studies. For example, the linear regression in Figure S1B has a positive y-intercept, and hence when the ATP unit (a.u.) is zero the absolute concentration is still nonzero. Although in the paper the ATP level is expressed as (a.u.), it is critical to know if this (a.u.) has absolute zero if we need to discuss about “fold change” of [ATP].

2. The transient increase of ATP level in late exponential phase has been reported [ref4, ref5]. The author should include these studies as reference. Regarding ATP levels in different carbon sources, the authors should also discuss the results in [ref6, ref 7].

3. I feel the biological meaning of ATP level need to be carefully interpreted, as many statements about ATP level regulation in this paper (e.g. line 147-148, line 153-154) can have alternative explanations. (a) In exponential phase, ATP level is the balance between ATP production and ATP consumption, and is the balance between nucleotide synthesis and nucleotide dilution by growth. (b) For non-steady-state, the situation is even more complex: Increases of ATP level could be resulted from slowing down of cell volume growth. It is known that during the entry of stationary phase *E. coli* cell sizes decreases, and this could also affect the ATP level. I suggest the authors to discuss these effects in Discussion.

4. In my opinion, whether a higher ATP level is beneficial for biosynthesis is still an open question. First, the ATP production flux may not be proportional to ATP level. The conversion between ATP and ADP can have a tunable turnover rate [ref 8], and a combination of high ATP level with low ATP turnover rate is possible. Second, ATP consumption flux may not be proportional to ATP level, since

physiological ATP level are much higher than K_d of many biosynthesis enzymes (i.e. ATP level is already saturated for these reactions) and hence enzymatic reactions can be robust to ATP level change.

On the other hand, there are studies reported strains with higher ATP level have enhanced protein production [ref 9]. Therefore, using ATP level as a proxy for metabolic burden is an interesting idea but requires further studies to support this concept.

5. Minor points:

(a) line 315: Here, Fig 4C has only three data points, one cannot infer “strong correlation” from this.

(b) line 325: Here, the ATP level only decrease from 16 to 14 (a.u.), hence one should not use the term “deplete” which suggest severe reduction of ATP level.

(c) line 367: exponential growth phase

Reference:

[1] <https://pubmed.ncbi.nlm.nih.gov/13701480/>

[2] <https://pubmed.ncbi.nlm.nih.gov/12081962/>

[3] <https://pubmed.ncbi.nlm.nih.gov/25283467/>

[4] <https://pubmed.ncbi.nlm.nih.gov/18805986/>

[5] <https://pubmed.ncbi.nlm.nih.gov/17965154/>

[6] <https://pubmed.ncbi.nlm.nih.gov/10660546/>

[7] <https://pubmed.ncbi.nlm.nih.gov/14670952/>

[8] <https://pubmed.ncbi.nlm.nih.gov/4554809/>

[9] <https://pubmed.ncbi.nlm.nih.gov/22173482/>

Dear reviewers,

Please note that the line numbers in this response correspond to the marked copy of the revised manuscript.

Reviewer #1:

Mu et al. use a previously developed ATP ratiometric sensor to quantify the ATP concentrations in *Escherichia coli* and *Pseudomonas putida* under different carbon conditions during different growth phases. Using the sensor, the authors corroborate previous findings that, in *E. coli*, acetate results in higher ATP levels than glucose. The authors find that in *P. putida*, oleate results in higher ATP levels.

The key engineering finding of this work is the demonstration that spiking acetate, in the case of *E. coli*, or oleate, in the case of *P. putida*, increases ATP levels resulting in higher fatty acid or PHA titers, respectively.

The key scientific finding of this manuscript is the discovery that ATP levels is a better metric than cell growth to determine reaction burden. This is shown when the authors discover that Module 3 of the limonene synthesis pathway shows similar ATP depletion levels than Modules 1 and 2 combined, even though Modules 1 and 2 use require 3 ATPs while Module 1 requires 1. The authors conclude by highlighting the potential of the ATP sensor for identifying bottlenecks in metabolic engineering.

This work will be of great significance to the metabolic engineering community as it shows that supplementation with often overlooked carbon sources can result in significantly higher titers. It would be interesting to determine if the other biosynthetic productivity metrics (yield, rates) also increase when increasing ATP levels. The work supports the conclusions in the manuscript. The methodology is sound and there is enough details in the methods for the work to be reproduced.

We appreciate the positive feedback from the reviewer and the suggestion to investigate additional bioproduction metrics. In response, we have quantified the overall yield and rate of FA production in *E. coli* and PHA production in *P. putida*, both with and without the supplementation of acetate or oleate, respectively. Our results revealed enhanced bioproduction yields and rates in both scenarios. We have added these detailed results as new Supplementary Figures S8 and S10 and discussed in lines 236, 292-293 of the manuscript.

Reviewer #2:

In this study, the authors measured cell growth and ATP level change with fluorescent biosensor iATPSnFR. The ATP level dynamics were measured in bulk by plate readers and growth dynamics were simultaneously measured by optical density.

The author reported several interesting findings about cellular ATP in *E. coli*, including (1) ATP level transiently increases during entry of stationary phase, (2) ATP level is higher in acetate media, and in glucose medium the amount of acetate supplement is positively correlated with fatty acid synthesis (3) ATP level is reduced in strains that synthesizing limonene. In addition, the author found in *P. putida* the oleate supplement increases cellular ATP level. Overall, the author proposed

to use ATP level as an indicator of metabolic burden, which can be examined for bioengineered strains that synthesize various kinds of compounds.

Response: We thank the reviewer for the positive review of our work and the insightful comments and references that helped us improve our manuscript.

Following are my comments:

1. There is one important technical issue in this work. The ATP concentration of *E. coli* measured by luciferase in this study (Fig. S1) is 10 to 100 times lower than the values reported in literatures, which is about several mM, or several micromolar per gram of cell dry weight [ref1, ref2, ref 3].

Related to this, the formula in line 445 for calculating [ATP] is confusing. The unit is incorrect, and I am wondering the term V_{sample} in the denominator should be a typo since the numerator already has [ATP] as concentration.

The calibration of ATP concentrations will affect the interpretations of some results in this studies. For example, the linear regression in Figure S1B has a positive y-intercept, and hence when the ATP unit (a.u.) is zero the absolute concentration is still nonzero. Although in the paper the ATP level is expressed as (a.u.), it is critical to know if this (a.u.) has absolute zero if we need to discuss about “fold change” of [ATP].

Response: We thank the reviewer for pointing out the mistake in our calculation. After examining, we realized that we had miscalculated the dilution factor when generating the standard curve from the luciferase assay. We have corrected this mistake and updated Fig. S1A in the Supplementary Information. After this correction, the ATP concentration as measured by luciferase assay varied from 1 to 5 mM, which falls into the same range as shown in Fig 2d of [ref3] as the reviewer suggested and Fig 2a of another reference [1] who reported *E. coli* ATP concentration in the unit of mM.

We also apologize for the confusion caused by the formula in the [ATP] calculation. The unit was correct; the formula has a V_{sample} in both the denominator and numerator to explain the calculation, and the V_{sample} can be canceled. To make it clear, we have modified the formula and provided further explanations.

$$[ATP]_{cell} = \frac{2.5[ATP]_{luc} \times V_{sample}}{n \times V_{sample} \times OD_{600} \times V_{cell}},$$

Specifically, on the numerator, $[ATP]_{luc}$ (in the unit of mM) is the ATP concentration measured from the luciferase assay. V_{sample} (in the unit of L) is the volume of each sample used for ATP quantification. The constant 2.5 is the dilution factor used in the luciferase assay. Thus, the numerator represents the total amount of ATP (in the unit of mmol) from each sample used in the luciferase assay. On the denominator, n is the number of cells per mL culture per OD, and $n \times V_{sample} \times OD_{600}$ is the total number of cells in each sample. V_{cell} (in the unit L) is the volume of a single *E. coli* cell. Thus, the denominator (in the unit of L) represents the total intracellular volume of each sample. $[ATP]_{cell}$ is the calculated intracellular ATP concentration in the unit of mM.

We appreciate the reviewer for pointing out the y-intercept in the linear regression. We found that setting the y-intercept to zero produces a better correlation coefficient ($R^2=0.95$, Fig S1B) than our

previous fitting with the y-intercept ($R^2=0.91$). Therefore, we used the new fitting without the y-intercept in Fig S1A. This does not change our other figures and conclusions.

Revised Fig. S1.

2. The transient increase of ATP level in late exponential phase has been reported [ref4, ref5]. The author should include these studies as reference. Regarding ATP levels in different carbon sources, the authors should also discuss the results in [ref6, ref 7].

Response: We sincerely thank the reviewer for providing these references for discussion. We have cited all these references and discussed results from these references in lines 363 to 369 of the revised manuscript.

Ref 7 particularly pointed out the problems in traditional ATP quantification methods: the results varied with ATP extraction methods. The ATP biosensor employed in this work does not require cell lysis and ATP extraction, thus offering advantages in ATP quantification.

3. I feel the biological meaning of ATP level need to be carefully interpreted, as many statements about ATP level regulation in this paper (e.g. line 147-148, line 153-154) can have alternative explanations. (a) In exponential phase, ATP level is the balance between ATP production and ATP consumption, and is the balance between nucleotide synthesis and nucleotide dilution by growth. (b) For non-steady-state, the situation is even more complex: Increases of ATP level could be resulted from slowing down of cell volume growth. It is known that during the entry of stationary phase *E. coli* cell sizes decreases, and this could also affect the ATP level. I suggest the authors to discuss these effects in Discussion.

Response: (a) We agree with the reviewer that the steady state ATP level is the balance between ATP production and ATP consumption, which is also what we claimed in lines 143-144 and in our model (see Supplementary Note 1). (b) We also agree with the reviewer's point that [ATP] at non-steady-state is more complex, and changes in cell volume could influence ATP concentration without affecting ATP abundance in each cell. To evaluate the impact of cell size changes on ATP

concentration, we measured our *E. coli* cell sizes during their transition to the stationary phase using microscopy. We found that the average cell volume change from the last round of cell division to the stationary phase is much smaller (<10%, Supplementary Fig S4, Supplementary Video) than the changes observed in ATP concentration. Therefore, we conclude that the changes in cell size are unlikely to be the major factor affecting ATP concentration.

We have added these data and discussions to lines 156 to 160.

4. In my opinion, whether a higher ATP level is beneficial for biosynthesis is still an open question. First, the ATP production flux may not be proportional to ATP level. The conversion between ATP and ADP can have a tunable turnover rate [ref 8], and a combination of high ATP level with low ATP turnover rate is possible. Second, ATP consumption flux may not be proportional to ATP level, since physiological ATP level are much higher than K_d of many biosynthesis enzymes (i.e. ATP level is already saturated for these reactions) and hence enzymatic reactions can be robust to ATP level change.

On the other hand, there are studies reported strains with higher ATP level have enhanced protein production [ref 9]. Therefore, using ATP level as a proxy for metabolic burden is an interesting idea but requires further studies to support this concept.

Response: We appreciate the reviewer for providing more critical insights into our study. We agree that ATP production is not always proportional to ATP level. From Equation (7) of our model in Supplementary Note 1:

$$[ATP]_{ss} = \frac{k_{prod}}{k_m} \frac{M(\mu)}{V(\mu)} - \frac{k_g}{k_m} \mu,$$

It is possible that the steady-state ATP level is high while the ATP synthesis rate (or turnover rate, k_{prod} in Equation 7) is low if the ATP consumption rate is also low (low k_g and high k_m). In this work, we observed that high FA productivity coincides with the ATP peak at the early stationary phase (Fig 1C, Fig 2B). Additionally, the higher FA production in acetate-supplemented media correlates with the higher ATP level under these conditions (Fig 2D-G). Although these observations do not lead to a causal relationship, the coincidence between high ATP levels and high FA production is obvious.

We also noticed that the typical ATP concentrations (measured using wild-type *E. coli* strain in sufficient nutrients) are higher than the K_d values of many ATP-using enzymes. However, this scenario may change in engineered strains or conditions when high ATP consumption lowers intracellular ATP levels, likely in the case of [ref 9] as the reviewer suggested, and in our conditions as well as in other references [2-4]. In these strains, a relatively higher ATP level is beneficial for the bioproduction of ATP-demanding products.

Metabolic burden can arise from multiple sources. We found that ATP dynamics are more sensitive in detecting metabolic burden from protein overexpression than simply measuring OD as previously used. We added further discussion (line 402-405) to point out the requirements of further studies on other types of metabolic burden.

5. Minor points:

(a) line 315: Here, Fig 4C has only three data points, one cannot infer “strong correlation” from this.

Response: We have revised our language by stating “both limonene titer and averaged ATP level decreased as the induction level increased”.

(b) line 325: Here, the ATP level only decrease from 16 to 14 (a.u.), hence one should not use the term “deplete” which suggest severe reduction of ATP level.

Response: Thank you for pointing this out. We revised the sentence by changing to “expression of the limonene pathway caused a notable decrease in cellular ATP levels without affecting growth”.

(c) line 367: exponential growth phase

Response: We have revised this sentence.

Reference

1. Deng, Y., et al., *Measuring and modeling energy and power consumption in living microbial cells with a synthetic ATP reporter*. BMC Biol, 2021. **19**(1): p. 101.
2. Tajima, Y., et al., *Impact of an energy-conserving strategy on succinate production under weak acidic and anaerobic conditions in Enterobacter aerogenes*. Microbial Cell Factories, 2015. **14**(1): p. 80.
3. Zhou, J., et al., *Improved ATP supply enhances acid tolerance of Candida glabrata during pyruvic acid production*. J Appl Microbiol, 2011. **110**(1): p. 44-534.
4. Singh, A., et al., *Manipulating redox and ATP balancing for improved production of succinate in E. coli*. Metabolic Engineering, 2011. **13**(1): p. 76-81.

REVIEWERS' COMMENTS

Reviewer #2 (Remarks to the Author):

The authors answered all my questions in the revision and I think the manuscript is ready to publish.